# Lentivirus Susceptibility in Brazilian and US Sheep with *TMEM154* Mutations

**DOI:** 10.3390/genes14010070

**Published:** 2022-12-26

**Authors:** Camila Souza Rodrigues, Danielle Assis de Faria, Thaísa Sant’Anna Lacerda, Samuel Rezende Paiva, Alexandre Rodrigues Caetano, Harvey Blackburn, Concepta McManus

**Affiliations:** 1Faculdade de Agronomia e Medicina Veterinária, Instituto Central de Ciências, Campus Darcy Ribeiro, Universidade de Brasília, Asa Norte, Brasilia 70910-900, DF, Brazil; 2Embrapa Recursos Genéticos e Biotecnologia, Final W5 Norte, Brasilia 70770-917, DF, Brazil; 3USDA-ARS—Agricultural Genetic Resources Preservation Research, 1111 South Mason Street, Fort Collins, CO 805214500, USA; 4Departamento de Ciências Fisiológicas, Instituto de Biologia, Campus Darcy Ribeiro, Universidade de Brasilia, Asa Norte, Brasilia 70910-900, DF, Brazil

**Keywords:** SNPs, *Ovis aries*, maedi-visna, Fluidigm, KASP

## Abstract

Small ruminant lentiviruses (SRLVs) affect sheep and goats worldwide. The major gene related to SRLV infections is the Transmembrane Protein Gene 154 (*TMEM154*). We estimated the haplotype frequencies of *TMEM154* in the USA (USDA-ARS) and Brazil (Embrapa) Gene Banks by using two different SNP genotyping methodologies, Fluidigm^TM^ and KASP^TM^. We also genotyped the *ZNF389_*ss748775100 deletion variant in Brazilian flocks. A total of 1040 blood samples and 112 semen samples from 15 Brazilian breeds were genotyped with Fluidigm for the SNP *ZNF389_*ss748775100 and 12 *TMEM154* SNPs. A total of 484 blood samples from the Santa Inês breed and 188 semen samples from 14 North American sheep breeds were genotyped with KASP for 6 *TMEM154* SNPs. All the Brazilian samples had the “I/I” genotype for the *ZNF389_*ss748775100 mutation. There were 25 *TMEM154* haplotypes distributed across the Brazilian breeds, and 4 haplotypes in the US breeds. Haplotypes associated with susceptibility were present in almost all breeds, which suggests that genetic testing can help to improve herd health and productivity by selecting non-susceptible animals as founders of the next generations. Fluidigm and KASP are reliable assays when compared with Beadchip arrays. Further studies are necessary to understand the unknown role of *TMEM154* mutations, host–pathogen interaction and new genes associated with the clinical condition.

## 1. Introduction

Maedi-visna virus (MVV) and caprine arthritis encephalitis virus (CAEV) are small ruminant lentiviruses (SRLVs) that belong to the genus *Lentivirus* in the *Retroviridae* family. They are distributed worldwide, causing significant economic losses in many countries [1]. Infections in sheep include interstitial pneumonia with dyspnea, indurative mastitis and cachexia, persisting for the lifetime of the host and causing chronic inflammation [2]. Seroconversion in MVV-infected sheep occurs over weeks to months, and, such as other diseases caused by lentiviruses, no effective treatment or vaccine is available [3]. Infected animals show progressive weight loss, low milk yield and a reduced production rate that can lead to premature culling [4].

Iceland was successful in ridding local sheep breeds of maedi-visna (MV) or Ovine Progressive Pneumonia (OPP) after an effort that lasted more than three decades [5]. There are no reports of MVV in Australia or New Zealand, despite problems with CAEV in goats from both countries [6,7]. Within the UK, MV has reduced lamb production by up to 40% in commercial flocks, whereas in the United States, studies have shown that nearly a quarter of the sheep herd is infected with the disease [8,9]. In Brazil, the condition is widespread in most of the states investigated, affecting flocks from the northeast [10,11,12,13] to the southeast region of the country [14,15,16,17].

Previous studies reported the occurrence of multiple candidate loci associated with a resistance/susceptibility to MV, such as *ZNF389* [18]*, CCR5* [19], *TLR9* [20], ovar*-DRB1* [21,22], *DPPA2/DPPA4* and *SYTL3* [23]. A previous study has reported that breed-specific differences can influence a higher or lower susceptibility to lentiviruses in small ruminants [24]. In a Genome-wide association study (GWAS), the deletion variant of *ZNF389* was linked with the proviral concentration of ovine Lentivirus (OvLV) found in some of the most common sheep breeds of the US [23]. Subsequently, another study indicated that insertion homozygotes had less than half the adjusted mean proviral concentration when compared with other genotypes [18].

Heaton et al. [25] identified a major gene related to OPP susceptibility, the transmembrane protein gene 154 (*TMEM154*). According to the authors, animals that have the ancestral haplotype (designated haplotype 3), which encodes glutamate (E) at position 35 and asparagine (N) at position 70, and the designated haplotype 2, which encodes isoleucine (I) at position 70 of the *TMEM* protein, are highly susceptible to MV. Conversely, those with haplotype 1, homozygous for the mutation encoding lysine (K) at position 35, are considered less susceptible to the infection.

After identifying the positive association between *TMEM154* and lentivirus susceptibility, a linkage disequilibrium (LD) between *TMEM154_E35K* locus and SNP *OAR17_5388531* (Illumina^®^ Ovine SNP50 Beadchip) was observed [26]. The LD between these two loci assign that the allele “*C*” at SNP *OAR17_5388531* is associated with a higher susceptibility to the infection. Other papers confirmed this association in sheep from Turkey [27], Germany [28], Iran [29] and Spain [30].

Although a link between the effects of *TMEM154* on MV is often reported, the role of some *TMEM154* mutations needs to be further investigated. Additionally, other studies are necessary to fully understand the impact of other co-receptors in the occurrence of the disease. The use of *TMEM154* as a proxy can make the genomic selection of animals with favorable haplotypes possible and help reduce the economic losses caused by MV [26].

The selection of animals with favorable haplotypes can also improve the quality of germplasm (e.g., semen and embryos) stored at gene banks. Ex situ collections, such as the Brazilian Animal Germplasm Biobank (BBGA) and the American germplasm collection managed by the US National Germplasm Program (NAGP), are valuable tools that, allied with in situ conservation, can help in the conservation of Animal Genetic Resources [31]. Therefore, genotyping samples kept at gene banks are crucial to guarantee the predominance of favorable haplotypes against MV. This can be an important strategy to optimize the quality of the material stored and ensure that genetic variability is maintained [32].

This study aimed at estimating the haplotype frequencies of the *TMEM154* gene and genotyping the *ZNF389_*ss748775100 deletion, related to lentivirus susceptibility, in the USA (USDA-ARS) and Brazil (Embrapa) animal genetic resources gene banks by using two different SNP genotyping methodologies, Fluidigm^TM^ and KASP^TM^.

## 2. Materials and Methods

### 2.1. Animal Samples 

The present study used 1040 blood samples from the sheep of 15 Brazilian breeds, kept in conservation centers of the Brazilian Agricultural Research Corporation (Embrapa) distributed across the country. The main criteria for sample selection were the biological material and breed availability in our National Gene Bank and the Embrapa Conservation Nucleus. Within breeds, the criterion was regional sampling. The breeds used in the study were: Santa Inês, Morada Nova, Crioula Lanada, Rabo Largo, Somali, Bergamasca, Corriedale, Ile de France, Pantaneiro, Dorper, Damara, Suffolk, Hampshire, Texel and Barriga Negra. Semen samples (N = 112) from these breeds kept at the Brazilian Animal Germplasm Biobank (BBGA) were also included in the study (Table 1). Additionally, we used 484 blood samples spotted on the FTA (Flinders Technology Associates) cards of the Santa Inês breed, belonging to a herd kept by a Conservation Nuclei at Embrapa Tabuleiros Costeiros (CPATC) and, finally, 188 animal samples of 14 North American breeds: Barbados Blackbelly, Black Welsh Mountain, Bluefaced Leicester, Hampshire, Hog Island, Katahdin, Leicester Longwool, Lincoln, Navajo Churro, Polypay, Rambouillet, Romanov, St. Croix and Suffolk kept at the National Center for Genetic Resources Preservation (NCGRP) (Table 1).

Semen samples were acquired from artificial insemination centers either by the American National Animal Germplasm Program or Brazilian Animal Germplasm Network as an effort to conserve genetic resources.

DNA extraction was performed using the Puregene purification protocol (Gentra Puregene^®^ Kit, QIAGEN, USA) for blood and semen samples. The DNA of the blood-spotted FTA^TM^ cards was extracted using the GenSolve™ DNA Recovery Kit (GenTegra, USA) according to the manufacturer’s instructions. A DNA quality check of the Brazilian sheep samples was performed in two ways. Initially, 1% agarose gel was used, stained with ethidium bromide, using lambda standards of 200 ng/µg, 100 ng/µg and 50 ng/µg for comparison. In addition, the samples were analyzed in a NanoDrop Thermo Scientific spectrophotometer (NanoDropTM 8000. Thermo Fisher Scientific, 2010. https://www.thermofisher.com/ accessed on 12 July 2015). The DNA obtained from the North American sheep was quantified by spectrophotometry only.

### 2.2. Genotyping Methodologies

We used two genotyping methodologies in the present study: *KASP™* (LGC Genomics, Hoddesdon, UK) and Fluidigm EP1™ system (Fluidigm, San Francisco, CA, USA). After a quality and quantity check, 90 µg of DNA was used for genotyping the Brazilian breeds, except for the Santa Inês samples from the CPATC. Genotyping data were generated according to standard protocols provided by Fluidigm for use with an EP1 platform. Each sample underwent pre-amplification with a set of Locus-Specific Primer (LSP) and Specific Target Amplification (STA) oligonucleotides for the assayed SNPs. Diluted amplified product was loaded into twelve 96.96 Dynamic Array™ IFCs (Fluidigm, San Francisco, CA, USA) with a ROX reference dye, amplification mixture, assays containing the Allele-Specific Primers (ASP) and LSP oligonucleotides for each SNP, according to the manufacturer’s instructions at Embrapa Genetic Resources and Biotechnology (Brasilia, Brazil). Briefly, Fluidigm^®^ SNP Type™ assays were designed for 12 selected SNPs with a Fluidigm D3™ assay web-based tool, according to the manufacturer’s instructions and rules described at [https://d3.fluidigm.com/account/login accessed on 12 July 2015]. Endpoint fluorescence image data were acquired on the EP1 Fluidigm imager, and the genotype calls were obtained by the Fluidigm SNP Genotyping Analysis Software.

The North American breeds and the Santa Inês samples from the CPATC were genotyped by KASP, using 10 ηg of DNA per sample in a 96-well plate format. The reaction and the components used in the KASP methodology are described at [http://www.lgcgenomics.com/genotyping/kasp-genotyping-reagents/ accessed on 12 July 2015]. Oligonucleotides were designed with the Kraken™ software system according to the manufacturer’s instructions (LGC Genomics, Hoddesdon, UK).

The SNPs *OAR17_*5388531 (rs414338245), *TMEM154_N70I* (rs427737740), *TMEM154_E35K* (rs408593969*), TMEM154_I102T, TMEM154_A13V-Fl2* and TMEM154_*T25I* were genotyped by both the KASP and Fluidigm methodologies. The SNP *ZNF389_*ss748775100 and SNPs *TMEM154_D33N (*rs429882112*), TMEM154_E31Q_v2, TMEM154_I74F* (rs410216979), *TMEM154_L14H, TMEM154_T44M* (rs42048963*0*) and *TMEM154_*E84Y were genotyped only by the Fluidigm methodology. The ZNF389 deletion variant NC_019477.1: g.29500068_29500069delAT ovine chromosome 20, NCBI dbSNP ss748775100) was genotyped with CGAATGGATCTTCAAGGCTTA and CAGCTTTTCCATGCAGAGTC as amplification primers, designed according to White et al. [18] and observing the specific rules of the Fluidigm assay. The flanking sequences of the *TMEM154* SNPs were designed according to Heaton et al. [26], observing each methodology’s rules which are provided in Appendix A.

Genotypes from the locus *OAR17_*5388531 within Illumina^®^ Ovine SNP50 and Illumina^®^ Ovine 600K were used to determine genotype reproducibility among methodologies. We were able to access 14 and 7 common animals between Fluidigm and Illumina^®^ Ovine SNP50 and Illumina^®^ Ovine 600K, respectively (Appendix A). Furthermore, 7 and 14 samples were used to compare KASP and Illumina^®^ Ovine SNP50 and Illumina^®^ Ovine 600K, respectively (Appendix A). Additionally, the locus *TMEM154_E35K* was accessed in 10 common animals between Fluidigm and Illumina^®^ Ovine 600K BeadChip (Appendix A). Finally, to compare both methodologies used in this study, we used 3 SNPs (*TMEM154_E35K*, *TMEM154_N70I* and *TMEM154_I102T)* in 27 samples (Appendix A). The Friedman test [33] was applied to determine whether the differences in the genotypes across the methodologies were statistically significant.

### 2.3. Data Analysis

Clustering was used to define the genotype classes for each SNP for both methods and was processed using GenomeStudio (Illumina Inc., San Diego, CA, USA). The confidence threshold for each genotype was >0.90 for both methodologies. Samples, in batches of 96, underwent quality control using SNP & Variation Suite v8.9.1—SVS (Golden Helix, Bozeman, MT, www.goldenhelix.com accessed on 15 November 2019) [34], eliminating samples with a call rate < 0.80 and markers with a call rate < 0.75. The linkage disequilibrium (LD) between the *TMEM154_E35K* and *OAR17_5388531* SNPs using r^2^ statistics [34] was also estimated using SVS v 8.9.1 (Golden Helix, Bozeman, MT, USA) [26]. Allele and haplotype frequencies were estimated by GenAlEx 6.5 [35] and Arlequin 3.5.2.2 [36]. Monomorphic SNPs were not included in the haplotype analysis. *Chi-square* test (*p* < 0.05) [37] was performed to determine whether the differences in allele frequencies across populations of the Santa Inês breed were statistically significant (Appendix A).

## 3. Results

### 3.1. Quality Control and Linkage Disequilibrium (LD)

All SNPs and samples passed the quality control filters for the KASP methodology. From the 6 SNPs genotyped, 3 were monomorphic in all breeds (*TMEM154_I102T, TMEM154_A13V* and *TMEM154_T25I*). After the quality control of the Fluidigm methodology, 845 samples with a call rate >0.80 remained. The SNPs *OAR17_5388531, TMEM154_A13V-Fl2, TMEM154_D33N, TMEM154_E31Q_v2, TMEM154_E35K, TMEM154_I74F, TMEM154_N70I, TMEM154_T44M* and *TMEM154_E84Y* successfully passed quality control. The SNPs *(TMEM154_L14H* and *TMEM154_T25I)* with a call rate <0.75 were excluded from the analysis. The SNP *TMEM154_I102T* was monomorphic, presenting the allele “T” in all breeds genotyped and was not included in the haplotype analysis. The locus *ZNF389_*ss748775100 was also monomorphic in all populations genotyped in this study, with all breeds homozygous for the “I/I” genotype. The r^2^ measure between the *TMEM154_E35K* and *OAR17_5388531* alleles across all populations were 0.96 and 0.93 on Fluidigm and KASP, respectively, indicating a strong LD between the allele “C” of the locus *OAR17_5388531* and the allele “G” of the SNP *TMEM154_E35K*.

### 3.2. Allele Frequency of Locus OAR17_5388531 in Brazilian and North American Sheep

The frequency of the ‘‘C’’ allele of the SNP *OAR17_5388531* ranged from 0.0% in the Hog Island breed to 100% in the Katahdin and Damara breeds (Figure 1). Breeds with a higher frequency of the “C” allele have higher frequencies of *TMEM154 E35* (highly susceptible haplotypes 2 and 3) and, consequently, are animals that are more prone to develop the disease. Alternatively, breeds with a low frequency of the C allele have a higher frequency of *TMEM154 K35* (less-susceptible haplotype 1), and, therefore, are animals that are less susceptible to OPP.

### 3.3. Distribution of TMEM154 Haplotypes in Brazilian and North American Sheep

The nine SNPs in the *TMEM154* gene (*OAR17_5388531, TMEM154_A13V-Fl2, TMEM154_D33N, TMEM154_E31Q_v2, TMEM154_E35K, TMEM154_I74F, TMEM154_N70I, TMEM154_T44M and TMEM154_E84Y)* generated 25 different haplotypes for the Brazilian sheep. The populations with the highest and lowest number of haplotypes were the BBGA and Damara, with 12 and 1, respectively. Among these haplotypes, the most common was ‘‘AQNETNFEI’’ (highly susceptible) (20%), occurring in 13 of the 16 breeds, followed by ‘‘AQDKTNFEI’’ (susceptible) (16%), which occurred in 13 breeds. The rarest haplotypes were “AQNETIFYΔI” (3%), found only in the Dorper breed, followed by ‘‘AQDKMNFEI’’ and AQNEMIIEI’’, both with a 1% frequency in the BBGA samples. Other rare haplotypes were ‘‘AQDEMNFEI’’ and ‘‘AQNKMNFEI’’, with a 1% frequency and present only in the Bergamasca breed. Furthermore, the haplotypes ‘‘AQNETNIYΔI’’ (1%) and ‘‘AQNEMNIYΔI’’ (1%) were unique to the Pantaneiro sheep. Finally, the haplotype ‘‘AQDKTIFYΔI’’ (1%) was detected only in the Rabo Largo breed, and the haplotypes ‘‘AQDETNIEI’’ (1%) and ‘‘AQDETIIEI’’ (1%), were exclusive to the Santa Inês (Table 2).

The 6 SNPs genotyped with the KASP methodology (*OAR17_5388531*, *TMEM154_N70I*, *TMEM154_E35K*, *TMEM154_I102T*, *TMEM154_A13V* and *TMEM154_T25I*) generated 4 haplotypes among the US breeds and the Santa Inês from the Embrapa Tabuleiros Costeiros (CPATC) (Table 3). The most common haplotype found was “KNIT” (less susceptible) (30%), present in 13 populations out of the 15 populations analyzed in this study, followed by “ENIT” (highly susceptible) (9%), present in 11 populations. The less common haplotype was “KIIT” (less susceptible) (3%), found only in the Bluefaced Leicester and Hampshire.

### 3.4. Data Comparison among Genotyping Platforms

The locus *OAR17_5388531* was genotyped on 14 common animals on both Fluidigm and Illumina^®^ Ovine SNP50. The reproducibility of the genotypes between methodologies was 100% (Appendix A), with all 28 alleles identical. The same result was obtained when we compared seven animals genotyped with Ilumina^®^ Ovine 600K BeadChip and Fluidigm (Appendix A). Another locus between Ilumina^®^ Ovine 600K BeadChip and Fluidigm was used to compare the repeatability of the genotypes between 10 common animals (*TMEM154_E35K*), and, once again, a total of 20 alleles were the same between the methodologies (Appendix A).

Regarding KASP and Illumina^®^ Ovine SNP50, we used seven samples and the locus *OAR17_5388531* as a reference. The results were the same as above, with a repeatability of 100% for the 14 alleles. For the same locus, using Ilumina^®^ Ovine 600K BeadChip and 14 samples (Appendix A), 28 alleles were identical.

Finally, comparing the two genotyping methodologies used here, KASP and Fluidigm, 27 common animals and 3 loci were analyzed: *TMEM154_E35K, TMEM154_N70I* and *TMEM154_I102T* (Appendix A). Following the pattern found in the previous analysis, the comparison between Fluidigm and KASP points to a repeatability of 100% on all 3 loci genotyped, as all 54 alleles were identical in the 27 common animals genotyped by both methodologies.

### 3.5. Differences in Allele Frequencies among Santa Inês Populations 

For some loci, a difference in the allele frequency was observed according to the geographic region of origin of the Santa Inês flock. The *TMEM154_N70I* locus have, in general, the allele “A” as the most frequent allele. However, the Empresa Baiana de Desenvolvimento Agricola (EBDA-BA) and the Embrapa Meio Norte-PI (CPAMN) populations have “T” as their highest frequency allele with frequencies of 0.75 and 0.63, respectively (χ^2^, *p* > 0.05) (Appendix A). Similar patterns occur for the locus *TMEM154_I74F* where the BBGA, CNPC and CPATC have the allele “A” as the most frequent. On the other hand, the UnB-DF and CPAMN presented the allele “T” as the most frequent (χ^2^, *p* < 0.05). Furthermore, the EBDA-BA has a frequency of 0.5 for each allele. For some loci, e.g., *TMEM154_A13V-Fl2* and *TMEM154_E31Q_v2*, we observed rare alleles with low frequencies in single populations: the CNPC (0.02, allele “T”) and CPAMN (0.04, allele “C”), respectively (Appendix A). Ultimately, the locus *TMEM154_D33N* has frequencies around 0.15 for the allele “G” in most Santa Inês populations (Appendix A). Still, for the UnB-DF and CNPC, the frequencies were 0.44 and 0.33, respectively (χ^2^, *p* < 0.05).

## 4. Discussion

The KASP method was 100% successful, as all designed SNPs were converted into primers and genotyped, and none of the SNPs or samples were eliminated by call rate in this methodology. For the Fluidigm methodology, we designed 13 assays and performed 12 different runs with 96.96 plates, in which each run has to be considered a separate event. Analysis of the entire dataset resulted in the elimination of 2 of the 13 SNPs (*TMEM154_L14H* and *TMEM154_T25I*) as they showed call rates <0.75 in 11 of the 12 plates genotyped. The quality filter for samples eliminated 307 samples with a call rate <0.80 in this methodology. This may be due to the high confidence threshold set up for each genotype, >0.90, increasing the accuracy of the results. It is important to mention that the SNP *TMEM154_I102T* was monomorphic in all breeds genotyped across both methodologies.

The data obtained from Fluidigm and KASP matched 100% with the results from the 50k and HD panels. Additionally, we did not find any differences in the genotypes between the two methodologies, indicating that both Fluidigm and KASP can be reliable alternatives for fast, customized and cost-effective genotyping. Furthermore, it is important to reinforce that the KASP methodology was 100% successful, with no samples or loci eliminated by call rate. However, samples and markers eliminated by call rate (<0.75) in the Fluidigm methodology can be explained by the higher confidence call, and each 96.96 plate genotyped represents a different PCR reaction, considered an independent genotyping event.

Haplotypes encoding (E) glutamate at position 35 of *TMEM154* are considered highly susceptible to OPP. In contrast, haplotypes encoding (K) lysine at the same position are considered less susceptible [25]. The high LD we found between the loci *OAR17_5388531* and *TMEM154_E35K* was expected and is in accordance with the results obtained by a previous study [26]. Mainly, we observed a higher frequency of the allele “C” in the breeds genotyped in this study. The highest frequency (100%) was found in the Damara, and the lowest (0.19%) in the Crioula Lanada. The Santa Inês breed is raised across Brazil. Nevertheless, we did not observe a difference in the allele frequencies for the locus *OAR17_5388531* among the populations, with a high frequency of the “C” allele in all populations analyzed. A high value of r^2^ indicates that the SNP *OAR17_5388531* can estimate highly susceptible *TMEM154* alleles in all the studied populations. *OAR17_5388531* can be used as a surrogate to relate the “C” allele with susceptibility, easing the genotyping process for breeders that have already performed it using Illumina^®^ Ovine SNP50 BeadChip or Illumina^®^ Ovine 600K BeadChip. Breed populations with a higher occurrence of the “C” allele are predicted to have a higher frequency of *K35* animals, which is associated with a lower susceptibility to the disease [26].

It is expected that distinct Santa Inês populations within the county present differences in allele frequencies, as demonstrated by previous authors with neutral markers [38]. The hypotheses can be raised that geographic distances affect the gene flow and level of admixture of the Santa Inês breed within certain populations [39,40]. For example, Santa Inês belonging to the UnB-DF herd seem to be crossbred with Bergamasca. This was proposed previously, where it was observed that Santa Inês animals from the center-west and southwest are genetically closer to Bergamasca than in other regions [38,39,40,41,42]. We did not find any significant allelic frequency differences among the BBGA, CNPC and CPATC Santa Inês populations. Two facts may explain this: (i) the BBGA is composed mainly of samples that come from the CNPC and CPATC flocks and (ii) both flocks are close geographically speaking, so there is an exchange of germplasm material between them. The same result could be expected for the Morada Nova breed. Despite the large number of animals of this breed, these are concentrated in the northeast region of the country [43], close to the CNPC conservation nuclei.

The selection of animals with favorable haplotypes is crucial to ensure the quality and the genetic variability of the germplasm stored in biobanks. Therefore, results such as those found here, where a single population of a breed has a rare allele or differences in allele frequencies, can aid in the long-term conservation of breed diversity and function as an important resource for breeders.

The *TMEM154_I102T* variant was observed for the first time in the Santa Inês breed, during resequencing by a previous study [26]. The authors identified the polymorphism in a single animal as a compound heterozygote for this locus. The animal was also heterozygous for the *TMEM154_N70I* variant. However, in our study, none of the populations presents heterozygous animals for the *TMEM154_I102T* locus. The *TMEM154_N70I* locus in the Santa Inês breed has a frequency of 0.6% for the allele “A” and 0.4% for the allele “T”. For the Santa Inês from the CPATC, the frequency was 0.7% for the allele “A” and 0.3% for the allele “T”, indicating heterozygous animals in the flocks. Overall, the allele “T” of the *TMEM154_N70I* locus has a total frequency of 5.9% in Brazilian sheep and 23.7% in North American sheep. Heaton et al. [26] called the variant *TMEM154_I102T* based on two reads of nine with high-quality scores, but, here, we could not reproduce the rare genotype found by the authors in any of the Brazilian breeds. We used this variant to test the reproducibility and check if this was an exclusive mutation of the Santa Inês breed.

Heaton et al. [26] defined haplotypes 1 (less susceptible), 2 (susceptible) and 3 (highly susceptible) in the Santa Inês breed. These authors also identified haplotypes 1 and 3 in the Crioula Lanada and Morada Nova breeds, as per the data obtained in the present study. Additionally, haplotype 6, considered rare, was previously observed in the Suffolk breed but not observed in the present study [26]. This might be due to the relatively low sample size used here. This reservation also applies to the following breeds: Bluefaced Leicester, Hog Island, Katahdin and Leicester Longwool, since their sample size is ≤2. Murphy et al. [44] confirmed that animals with diplotype “1”, “1” of the *K35* variant of *TMEM154* had a reduced incidence of OPP infection, which leads to an improvement in productivity. Furthermore, the author suggests that the selection of sheep with the *TMEM154* haplotype “1” in flocks with a high frequency of haplotype “3”, can be a cost-effective alternative to reduce the economic damage caused by OPP. However, further investigation is required to understand the effect of other *TMEM154* mutations on the susceptibility to the disease and whether these effects can be extended to multiple breeds.

SRLVs present high genetic variability, contributing to the evolution of multiple viral strains worldwide [45]. MVV-like and CAEV-like strains of SRLVs were primarily considered strictly host-specific for sheep and goats, respectively. However, recent investigations indicated that cross-species transmission events are possible in sheep and goats from Brazilian mixed flocks [46,47]. Ramirez et al. [30] suggest that selection based on *TMEM154* is suitable for specific SRLVs strains and ovine breeds. Nonetheless, the same authors propose that generalization to the whole genetic spectrum of Lentiviruses, ovine breeds and epidemiological situation across the globe may need further validation.

Molaee et al. [29] indicate that the association of the SNP *TMEM154_E35K* with a susceptibility to the disease must be undertaken carefully as, in the Merinoland breed, a high number of KK (less-susceptible genotype) sheep was positive for SRLV. Therefore, follow-up breed-specific studies can be useful to detect new *TMEM154* variants associated with the development of MV, as well as to understand the unknown role of already identified markers. Furthermore, other genes such as Ovine-*DRB1* can be associated with a susceptibility to MV, implying that studies aimed at understanding other genetic factors involved in the occurrence of SRLVs are crucial [48].

White et al. [18] established an association between the deletion variant *ZNF389_*ss748775100 and higher proviral concentrations in Rambouillet, Polypay and Columbia sheep from Idaho, US. Conversely, in our study, no breeds genotyped presented the deletion allele, as all populations were homozygous for the insertion “I/I”. According to White et al. [18], animals with this genotype had less than half the adjusted mean proviral concentrations, which could possibly indicate an association with susceptibility to the virus.

Although *ZNF389* has been previously associated with MV, further investigation is required to understand the functional importance of this genomic region. Zinc finger proteins (ZFPs) are characteristic DNA binding domains that can be found in a variety of transcription factors [49]. White et al. [18] implicate that *ZNF389* plays a biological role related to ovine lentivirus proviral concentration in sheep. The same authors propose that one or more zinc finger genes located in this region can act as a transcriptional regulator of host genes, such as *TRIM5a*, limiting the proviral replication of the virus [18,50].

The results obtained in this study can be of paramount importance for the Brazilian and American gene banks. For instance, investigation of the *ZNF389_*ss748775100 deletion variant genotypes, the *TMEM154* haplotypes and their frequencies in the breeds can help to select animals with favorable alleles and haplotypes. Nevertheless, gene banks should store all genotypes to facilitate future work on susceptibility and resistance. Consequently, these methodologies can be reliable resources for breeders if MV becomes a threat, thus aiding in the conservation of breed diversity.

## 5. Conclusions

Overall, haplotypes associated with a lower susceptibility risk to OPP are common in both Brazilian and North American sheep. This suggests an opportunity to reduce Lentivirus susceptibility in multiple sheep breeds using genomic selection. There is a significant difference in the allele frequencies of the *TMEM154* mutations among different Santa Inês populations, resulting from the genetic subdivision previously observed in the breed. Contrary to the insertion allele, the *ZNF389_*ss748775100 deletion variant is associated with higher proviral concentrations in sheep. However, we did not detect the deletion variant in any of the breeds here.

Genetic variability is crucial to ensure the quality of the material stored in conservation centers and gene banks. The results found here of the *TMEM154* haplotypes and their frequency suggest a range of genetic variability has been captured by the gene banks. A few breeds showed a lack of variability, which was likely due to a small sample size. Finally, the comparison of the results obtained from the Fluidigm and KASP assays with more robust technologies, such as the 50k and the HD panels, indicate that these methodologies are reliable. Therefore, they can be useful, cost-effective tools to improve genomic selection programs. Further validation is still necessary to understand the unknown role of some of the *TMEM154* mutations and the role of other genetic factors associated with the disease.

## Figures and Tables

**Figure 1 genes-14-00070-f001:**
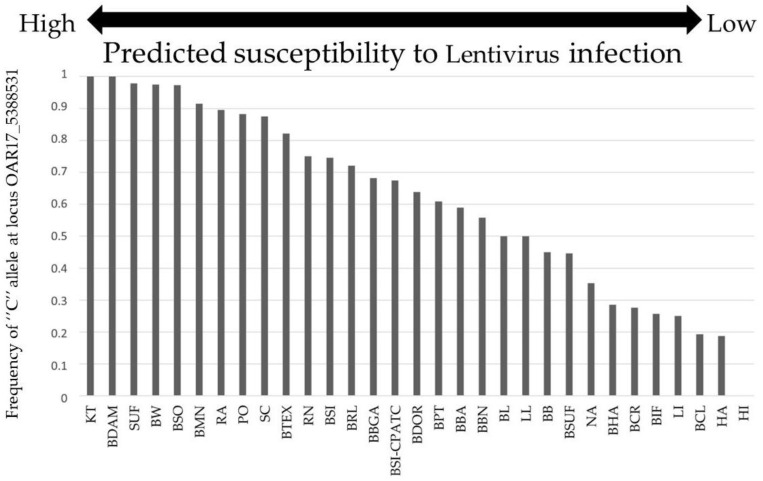
Frequency of the allele “C” at the locus *OAR17_5388531* in Brazilian and North American sheep breeds. The “C” allele is in LD with the “G” allele in the codon 35 (GAA) of the *TMEM154* gene. KT—Katahdin; BDAM—Brazilian Damara; SUF—Suffolk; BW—Black Welsh Mountain; BSO—Brazilian Somali; BMN—Brazilian Morada Nova; RA—Rambouillet; PO—Polypay; SC—St. Croix; BTEX—Brazilian Texel; RN—Romanov; BSI—Brazilian Santa Inês; BRL—Brazilian Rabo Largo; BBGA—Brazilian Animal Germplasm Biobank; BSI-CPATC – Brazilian Santa Inês-CPATC; BDOR—Brazilian Dorper; BPT—Brazilian Pantaneiro; BBA—Brazilian Bergamasca; BBN—Brazilian Barriga Negra; BL—Bluefaced Leicester; LL—Leicester Longwool; BB—Barbados Blackbelly; BSUF—Brazilian Suffolk; NA—Navajo Churro; BHA—Brazilian Hampshire; BCR—Brazilian Corriedale; BIF—Brazilian Ile de France; LI—Lincoln; BCL—Brazilian Crioula Lanada; HA—Hampshire; HI—Hog Island.

**Table 1 genes-14-00070-t001:** Methodology and number of samples used in the present study according to sheep breed and country of origin.

Breed Code	Breed/Group	Country	*n*	Genotyping Methodology
BBGA	Brazilian Animal Germplasm Biobank	Brazil	88	Fluidigm
BBN	Brazilian Barriga Negra	Brazil	43	Fluidigm
BBA	Brazilian Bergamasca	Brazil	39	Fluidigm
BCR	Brazilian Corriedale	Brazil	29	Fluidigm
BCL	Brazilian Crioula Lanada	Brazil	44	Fluidigm
BDAM	Brazilian Damara	Brazil	10	Fluidigm
BDOR	Brazilian Dorper	Brazil	18	Fluidigm
BHA	Brazilian Hampshire	Brazil	14	Fluidigm
BIF	Brazilian Ile de France	Brazil	39	Fluidigm
BMN	Brazilian Morada Nova	Brazil	129	Fluidigm
BPT	Brazilian Pantaneiro	Brazil	46	Fluidigm
BRL	Brazilian Rabo Largo	Brazil	34	Fluidigm
BSI	Brazilian Santa Inês	Brazil	233	Fluidigm
BSO	Brazilian Somali	Brazil	37	Fluidigm
BSUF	Brazilian Suffolk	Brazil	28	Fluidigm
BTEX	Brazilian Texel	Brazil	14	Fluidigm
BSI	Brazilian Santa Inês-CPATC	Brazil	484	KASP
BB	Barbados Blackbelly	USA	10	KASP
BW	Black Welsh Mountain	USA	20	KASP
BL	Bluefaced Leicester	USA	1	KASP
HA	Hampshire	USA	8	KASP
HI	Hog Island	USA	1	KASP
KT	Katahdin	USA	1	KASP
LL	Leicester Longwool	USA	2	KASP
LI	Lincoln	USA	8	KASP
NA	Navajo Churro	USA	17	KASP
PO	Polypay	USA	17	KASP
RA	Rambouillet	USA	70	KASP
RN	Romanov	USA	8	KASP
SC	St. Croix	USA	23	KASP
SUF	Suffolk	USA	2	KASP

**Table 2 genes-14-00070-t002:** *TMEM154* haplotype frequencies of Brazilian sheep generated by 9 SNPs with the Fluidigm methodology.

Breed/Group ^1^	H1 ^2^	H2	H3	H4	H5	H6	H7	H8	H9	H10	H11	H12	H13	H14	H15	H16	H17	H18	H19	H20	H21	H22	H23	H24	H25	Missing Data
BBGA	0.17	0.063	0.011	0.011	0.33	0.119	0.091	0.011	0.017	0.011	0.006	0.011														0.148
BBN	0.163	0.14			0.209	0.163	0.058	0.023	0.023				0.012	0.093												0.116
BBA	0.028	0.216			0.088	0.13		0.115							0.038	0.013	0.013	0.026								0.333
BCR	0.247	0.424			0.01	0.128		0.034		0.034																0.122
BCL	0.591	0.034			0.034		0.102	0.011																		0.227
BDAM						0.8																				0.2
BDOR		0.222				0.278		0.111											0.028	0.028						0.333
BHA	0.07	0.25	0.071		0.04				0.04	0.04																0.489
BIF	0.06	0.244				0.103		0.026							0.026											0.543
BMN	0.03	0.007			0.786	0.008	0.039							0.023												0.106
BPT	0.391				0.315		0.087		0.12												0.011	0.011				0.065
BRL	0.147	0.115			0.103	0.252		0.027															0.011			0.344
BSI	0.115	0.029			0.21	0.076	0.173	0.057				0.006	0.011		0.018									0.008	0.008	0.29
BSO					0.014	0.473	0.014	0.162											0.014							0.324
BSUF	0.357	0.25	0.036			0.054		0.089		0.054								0.018								0.143
BTEX		0.143				0.107		0.107		0.036								0.036								0.571

^1^ Breeds/Groups: BBGA—Brazilian Animal Germplasm Bank; BBN—Brazilian Barriga Negra; BBA—Brazilian Bergamasca; BCR—Brazilian Corriedale; BCL—Brazilian Crioula Lanada; BDAM—Brazilian Damara; BDOR—Brazilian Dorper; BHA—Brazilian Hampshire; BIF—Brazilian Ile de France; BMN—Brazilian Morada Nova; BPT—Brazilian Pantaneiro; BRL—Brazilian Rabo Largo; BSI—Brazilian Santa Inês; BSO—Brazilian Somali; BSUF—Brazilian Suffolk; BTEX—Brazilian Texel. ^2^ Haplogroups: HN—AQNKTNIEI; HO—AQDKTIFEI; HP—AQDEMNFEI; HQ—AQNKMNFEI; HR—AQNEMIFEI; HS—AQNETNFYΔI—HT—AQNETIFYΔI; HU—AQNETNIYΔI; HV—AQNEMNIYΔI; HW—AQDKTIFYΔI; HX—AQDETNIEI; HY—AQDETIIEI.

**Table 3 genes-14-00070-t003:** *TMEM154* haplotype frequencies of North American breeds and Brazilian Santa Inês generated by 6 SNPs with the KASP methodology.

Breed ^1^	H1 ^2^	H2	H3	H4
BB	0.900	0.100		
BW	0.600	0.200	0.200	
BL		0.500		0.500
HA	0.125		0.625	0.250
HI	1.000			
KT	1.000			
LL	1.000			
LI	1.000			
NA	0.852	0.058	0.088	
PO	0.970	0.029		
RA	0.602	0.191	0.191	
RN		1.000		
SC	0.562	0.125	0.187	
SUF	0.608	0.086	0.304	
BSI (CPATC)	0.149	0.518	0.275	

^1^ Breeds: BB—Barbados Blackbelly; BW—Black Welsh Mountain; BL—Bluefaced Leicester; HA—Hampshire; HI—Hog Island; KT—Katahdin; LL—Leicester Longwool; LI—Lincoln; NA—Navajo Churro; PO—Polypay; RA—Rambouillet; RN—Romanov; SC—St. Croix; SUF—Suffolk; BSI—Brazilian Santa Inês (CPATC). ^2^ Haplogroups: H1—K N I T; H2—E N I T; H3—E I I T; H4—K I I T.

## Data Availability

Data available on request due to restrictions.

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
