# Peer review of "Lentivirus Susceptibility in Brazilian and US Sheep with *TMEM154* Mutations"

_genes, 2022, doi:10.3390/genes14010070_

Round 1
Reviewer 1 Report
Authors report on frequencies of TMEM154 in the USA (USDA-ARS) and Brazil (Embrapa) Gene Banks by two different SNP genotyping methodologies, Fluidigm and KASP. They also genotyped the ZNF389_ss748775100 deletion variant in Brazilian flocks. The results indicate that by selecting non-susceptible animals as founders of the next generations will improve herd health and productivity. This has certain guiding significance to sheep production. But there are still some problems that authors need to be clarified.
1. I suggest that the first and second paragraphs in the introduction section should be merged into one paragraph, because they are mainly talking about the harm of SRLVs-infected sheep.
2. I suggest that the third and fourth paragraphs in the introduction section should be combined into one paragraph, because they are all mainly discussing CAEV or MVV infection in different countries.
3. The report used two genotyping methods: Fluidigm and KASP. So why did Brazilian Santa Inês-CPATC and North American breeds choose KASP?
4. I suspect that the sample number for partial breeds (e.g Bluefaced Leicester, Hog Island, Katahdin, Leicester Longwool and Suffolk) are less than 2, the authors should consider whether the sample size is sufficient and the results are convincing.
5. Line 380: 'indicate' should be 'indicated'.
Author Response
New manuscript version attached (genes-2045873_221212.docx)
- Suggestion accepted and paragraphs 1 and 2 were merged.
- Suggestion accepted and paragraphs 3 and 4 were merged.
- The KASP methodology was selected previously and used in our ‘first study’ which included only USA samples and Santa Inês-CPATC samples. It was before Fluidigim methodology was implemented in our Research Center in Brazil. In this way, our ‘second’ study were carried out using the Fluidigim technology with a broader sampling from Brazil. As most of the markers were the same, the goals of both studies were also the same and the authors shared the opinion of a combined publication, therefore with two distinct methodologies.
- One sentence about those breeds with sample size ≤ 2 was added to the discussion section (new document lines 389-390).
- Spelling was corrected at line 401.

Reviewer 2 Report
The present research talks about “Estimation of haplotype frequencies of the TMEM154 and gen-86 gene typing the ZNF389_ss748775100 deletion, related to lentivirus susceptibility, in the US Animal Genetic Resources gene banks 87 (USDA-ARS) and Brazil (Embrapa) by two 88 different SNP genotyping methodologies, Fluidigm and KASP”. This research provides interesting information. However, it is necessary to make some changes.
MATERIAL AND METHODS
Line 91.- What animal welfare guidelines and standards did you use? Do you have an ethics committee that supports the research?
Line 93.- I recommend mentioning the criteria for the selection of the animals that were sampled, such as age, sex, etc. In addition, it would be convenient to mention the average weight and body condition.
Line 107-108.- mentions "Gentra 107 Puregene® Kit) for blood and semen samples" while line 93 mentions "1040 blood samples", clearly define the origin of the samples.
RESULTS
General comments: each variable described in results must be described in the materials and methods section.
Line 181.- I recommend restructuring this section “3.1. Quality Control and Linkage Disequilibrium (LD)" because they mention methodological issues which would be better described in materials and methods.
Line 205.- figures 1, the literals are very small and difficult to understand. I recommend increasing it
Line 217.- I recommend passing this sentence “In contrast, haplotypes encoding (K) lysine at the same position are considered less susceptible [25].” To the discussion section.
DISCUSSION
General comments:
I recommend that the discussion be in agreement as the results were reported. I mean, Frequency of allele "C, Distribution of TMEM154 haplotypes, etc.
Line 94-313.- I suggest changing this paragraph to the results section because only results are reported and they are not discussed or contrasted with other investigations.
CONCLUSION
I recommend that it be more concise. In addition to mentioning their more about its possible implications and importance of this research.
Author Response
New manuscript version attached (genes-2045873_221212.docx)
MATERIAL and METHODS
Line 91 – The animal Care and Use Committee Guidelines of Embrapa ethics committee number was added (new manuscript main document, lines 471-473).
Line 93 – We have adjusted the text to better explain the sampling criteria (new manuscript main document, lines 100-102). All animals alive kept in conservation centers of the Brazilian Agricultural Research Corporation (Embrapa) distributed across the country were genotyped. Also the Brazilian Animal Germplasm Biobank (BBGA) and the American Germplasm Bank were genotyped, in this case, the BBGA was also genotyped completely with no selection of samples. Due to its size the American Germplasm Bank could not be genotyped fully, samples were picked randomly or form different regions. Speaking of average weight and body condition for most of the animals we don´t have this information. Some animals are not alive; some are not in the herd anymore, especially those kept at gene banks. Again, the idea o f the manuscript is increase information for the stored samples even the ones with or without phenotypic data.
Line 107-108: The same extraction kit (Gentra Puregene® Kit) was used for both semen and blood samples. As specified at line 98 (new manuscript main document), 1040 blood samples were used; followed by line 105 (new manuscript main document), 112 semen samples. A total of 1152 samples were subject of DNA extraction using Gentra Puregene® Kit. Furthermore, line 109 (new manuscript main document), 188 samples kept at the National Center for Genetic Resources Preservation had also their DNA extracted from semen using the same kit mentioned above. The authors explained in the first paragraph and Table 1 the tissue of origin of each sample and therefore the extraction kit used.
RESULTS
General comments: We did not find any variables present at the results which were not cited in Material and Methods. Although we have added some sentences to M&M section in order to better clarification of item 2.3. Data Analysis (new manuscript main document, line 180 and 186-187).
Line 181 – We carefully reanalyzed the section 3.1 and we suggest if this section could stay as it is. The section describes the results obtained by the quality control filters on both methodologies, which samples and markers were eliminated based on that. Furthermore, the paragraph brings results of the LD analysis. Values of r² obtained between the markers in question on both methodologies as well. The criteria used for quality control filters and the statistic and software used to estimate LD were described in M&M.
Line 205 – suggestion accepted. The figure was changed and its caption was improved for better understanding (new manuscript main document, lines 215-230).
Line 217 – Suggestion accepted. The sentence was withdrawn from M&M and allocated at discussion section (new manuscript main document, lines 336-338).
DISCUSSION
General comments: we did a revision in the discussion section with small amendments.
Line 94-313 – If we understood correct, this is the paragraph about the results from Fluididigm and KASP comparing with 50 K and HD panels (section 3.4 of the results). There is a brief discussion raising hypotheses to explain the results observed at discussion about this subject (new manuscript main document, lines 327-335). There is no citation here because to the best of our knowledge, there is no other study that performed such comparison.
CONCLUSION
Suggestion accepted. The conclusion section was modified and reduced as requested.
